# DNA-GAN: Learning Disentangled Representations from Multi-Attribute Images

## Abstract

Disentangling factors of variation has always been a challenging problem in representation learning. Existing algorithms suffer from many limitations, such as unpredictable disentangling factors, bad quality of generated images from encodings, lack of identity information, etc. In this paper, we proposed a supervised algorithm called DNA-GAN trying to disentangle different attributes of images. The latent representations of images are DNA-like, in which each individual piece represents an independent factor of variation. By annihilating the recessive piece and swapping a certain piece of two latent representations, we obtain another two different representations which could be decoded into images. In order to obtain realistic images and also disentangled representations, we introduced the discriminator for adversarial training. Experiments on Multi-PIE and CelebA datasets demonstrate the effectiveness of our method and the advantage of overcoming limitations existing in other methods.

## 1 Introduction

The success of machine learning algorithms depends on data representation, because different representations can entangle different explanatory factors of variation behind the data. Although prior knowledge can help us design representations, the vast demand of AI algorithms in various domains cannot be met, since feature engineering is labor-intensive and needs domain expert knowledge. Therefore, algorithms that can automatically learn good representations of data will definitely make it easier for people to extract useful information when building classifiers or predictors.

Of all criteria of learning good representations as discussed in Bengio et al. (2013), disentangling factors of variation is an important one that helps separate various explanatory factors. For example, given a human-face image, we can obtain various information about the person, including gender, hair style, facial expression, with/without eyeglasses and so on. All of these information are entangled in a single image, which renders the difficulty of training a single classifier to handle different facial attributes. If we could obtain a disentangled representation of the face image, we may build up only one classifier for multiple attributes.

In this paper, we propose a supervised method called DNA-GAN to obtain disentangled representations of images. The idea of DNA-GAN is motivated by the DNA double helix structure, in which different kinds of traits are encoded in different DNA pieces. We make a similar assumption that different visual attributes in an image are controlled by different pieces of encodings in its latent representations. In DNA-GAN, an encoder is used to encode an image to the attribute-relevant part and the attribute-irrelevant part, where different pieces in the attribute-relevant part encode information of different attributes, and the attribute-irrelevant part encodes other information. For example, given a facial image, we are trying to obtain a latent representation that each individual part controls different attributes, such as hairstyles, genders, expressions and so on. Though annihilating recessive pieces and swapping certain pieces, we can obtain novel crossbreeds that can be decoded into new images. By the adversarial discriminator loss and the reconstruction loss, DNA-GAN can reconstruct the input images and generate new images with new attributes. Each attribute is disentangled from others gradually though iterative training. Finally, we are able to obtain disentangled representations in the latent representations.

The summary of contributions of our work is as follows:

1. We propose a supervised algorithm called DNA-GAN, that is able to disentangle multiple attributes as demonstrated by the experiments of interpolating multiple attributes on Multi-PIE (Gross et al., 2010) and CelebA (Liu et al., 2015) datasets.

2. We introduce the annihilating operation that prevents from trivial solutions: the attribute-relevant part encodes information of the whole image instead of a certain attribute.

3. We employ iterative training to address the problem of unbalanced multi-attribute image data, which was theoretically proved to be more efficient than random image pairs.

## 2   RELATED WORK

Traditional representation learning algorithms focus on (1) probabilistic graphical models, characterized by Restricted Boltzmann Machine (RBM) (Smolensky, 1986), Autoencoder (AE) and their variants; (2) manifold learning and geometrical approaches, such as Principal Components Analysis (PCA) (Pearson, 1901), Locally Linear Embedding (LLE) (Roweis & Saul, 2000), Local Coordinate Coding (LCC) (Yu et al., 2009), etc. However, recent research has actively focused on developing deep probabilistic models that learn to represent the distribution of data. Kingma & Welling (2013) employs an explicit model distribution and uses variational inference to learn its parameters. As the generative adversarial networks (GAN) (Goodfellow et al., 2014) has been invented, many implicit models are developed.

In the semi-supervised setting, Siddharth et al. (2016) learns a disentangled representations by using an auxiliary variable. Bouchacourt et al. (2017) proposes the ML-VAE that can learn disentangled representations from a set of grouped observations. In the unsupervised setting, InfoGAN (Chen et al., 2016) tries to maximize mutual information between a small subset of latent variables and observations by introducing an auxiliary network to approximate the posterior. However, it relies much on the a-priori choice of distributions and suffered from unstable training. Another popular unsupervised method $\beta$-VAE (Higgins et al., 2016), adapted from VAE, lays great stress on the KL distance between the approximate posterior and the prior. However, unsupervised approaches do not anchor a specific meaning into the disentanglement.

More closely with our method, supervised methods take the advantage of labeled data and try to disentangle the factors as expected. DC-IGN (Kulkarni et al., 2015) asks the active attribute to explain certain factor of variation by feeding the other attributes by the average in a mini-batch. TD-GAN (Wang et al., 2017) uses a tag mapping net to boost the quality of disentangled representations, which are consistent with the representations extracted from images through the disentangling network. Besides, the quality of generated images is improved by implementing the adversarial training strategy. However, the identity information should be labeled so as to preserve the id information when swapping attributes, which renders the limitation of applying it into many other datasets without id labels. IcGAN (Perarnau et al., 2016) is a multi-stage training algorithm that first takes the advantage of cGAN (Mirza & Osindero, 2014) to learn a map from latent representations and conditional information to real images, and then learn its inverse map from images to the latent representations and conditions in a supervised manner. The overall effect depends on each training stage, therefore it is hard to obtain satisfying images. Unlike these models, our model requires neither explicit id information in labels nor multi-stage training.

Many works have studied the image-to-image translation between unpaired image data using GAN-based architectures, see Isola et al. (2016), Taigman et al. (2016), Zhu et al. (2017), Liu et al. (2017) and Zhou et al. (2017). Interestingly, these models require a form of 0/1 weak supervision that is similar to our setting. However, they are circumscribed in two image domains which are opposite to each other with respect to a single attribute. Our model differs from theirs as we generalize to the case of multi-attribute image data. Specifically, we employ the strategy of iterative training to overcome the difficulty of training on unbalanced multi-attribute image datasets.

## 3   DNA-GAN APPROACH

In this section, we formally outline our method. A set $\mathscr{X}$ of multi-labeled images and a set of labels $\mathscr{Y}$ are considered in our setting. Let $\{(\mathbf{X}^1, \mathbf{Y}^1), \ldots, (\mathbf{X}^m, \mathbf{Y}^m)\}$ denote the whole training dataset, where $\mathbf{X}^i \in \mathscr{X}$ is the $i$-th image with its label $\mathbf{Y}^i \in \mathscr{Y}$. The small letter $m$ denotes the number

of samples in set $\mathscr{X}$ and $n$ denotes the number of attributes. The label $\mathbf{Y}^i = (\mathbf{y}_1^i, \ldots, \mathbf{y}_n^i)$ is a $n$-dimensional vector where each element represents whether $\mathbf{X}^i$ has certain attribute or not. For example, in the case of labels with three candidates [Bangs, Eyeglasses, Smiling], the facial image $\mathbf{X}^i$ whose label is $\mathbf{Y}^i = (1, 0, 1)$ should depict a smiling face with bangs and no eyeglasses.

## 3.1 MODEL

As shown in Figure 1, DNA-GAN is mainly composed of three parts: an encoder (Enc), a decoder (Dec) and a discriminator (D). The encoder maps the real-world images $A$ and $B$ into two latent disentangled representations

$$\text{Enc}(A) = [a_1, \ldots, a_i, \ldots, a_n, z_a], \quad \text{Enc}(B) = [b_1, \ldots, b_i, \ldots, b_n, z_b] \tag{1}$$

where $[a_1, \ldots, a_i, \ldots, a_n]$ is called the attribute-relevant part, and $z_a$ is called the attribute-irrelevant part. $a_i$ is supposed to be a DNA piece that controls $\mathbf{y}_i$, the $i$-th attribute in the label, and $z_a$ is for keeping other silent factors which do not appear in the attribute list as well as image identity information. The same thing applies for $\text{Enc}(B)$.

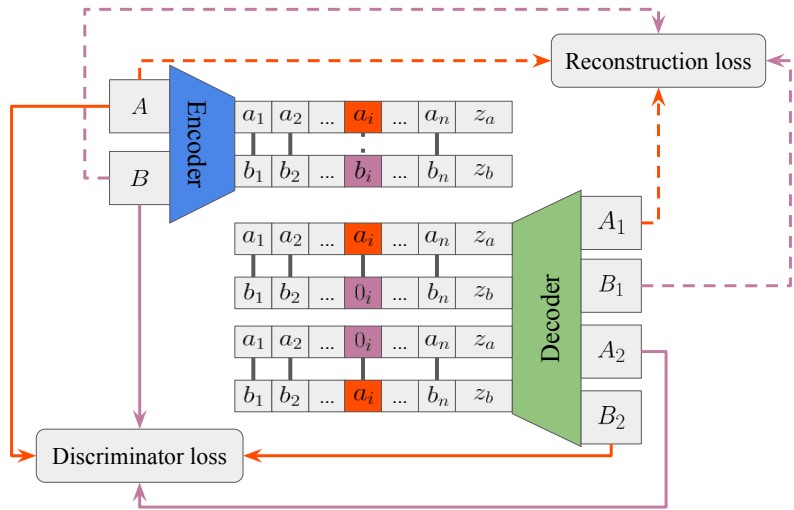

Figure 1: DNA-GAN architecture.

We focus on one attribute each time in our framework. Let's say we are at $i$-th attribute. $A$ and $B$ are required to have different labels, i.e. $(\mathbf{y}_1^A, \ldots, 1_i^A, \ldots, \mathbf{y}_n^A)$ and $(\mathbf{y}_1^B, \ldots, 0_i^B, \ldots, \mathbf{y}_n^B)$, respectively. In our convention, $A$ is always for the dominant pattern, while $B$ is for the recessive pattern. We copy $\text{Enc}(A)$ directly as the latent representation of $A_1$, and annihilate $b_i$ in the copy of $\text{Enc}(B)$ as the latent representation of $B_1$. The annihilating operation means replacing all elements with zeros, and plays a key role in disentangling the attribute, which we will discuss in detail in Section 3.3. By swapping $a_i$ and $0_i$, we obtain two new latent representations $[a_1, \ldots, 0_i, \ldots, a_n, z_a]$ and $[b_1, \ldots, a_i, \ldots, b_n, z_b]$ that are supposed to be decoded into $A_2$ and $B_2$, respectively. Though a decoder Dec, we can get four newly generated images $A_1$, $B_1$, $A_2$ and $B_2$.

$$\begin{aligned} \text{Dec}([a_1, \ldots, a_i, \ldots, a_n, z_a]) = A_1, \quad \text{Dec}([b_1, \ldots, 0_i, \ldots, b_n, z_b]) = B_1 \\ \text{Dec}([a_1, \ldots, 0_i, \ldots, a_n, z_a]) = A_2, \quad \text{Dec}([b_1, \ldots, a_i, \ldots, b_n, z_b]) = B_2 \end{aligned} \tag{2}$$

Out of these four children, $A_1$ and $B_1$ are reconstructions of $A$ and $B$, while $A_2$ and $B_2$ are novel crossbreeds. The reconstruction losses between $A$ and $A_1$, $B$ and $B_1$ ensure the quality of reconstructed samples. Besides, using an adversarial discriminator D that helps make generated samples $A_2$ indistinguishable from $B$, and $B_2$ indistinguishable from $A$, we can enforce attribute-related information to be encoded in $a_i$.

## 3.2 Loss Functions

Given two images $A$ and $B$ and their labels $\mathbf{Y}^A = (\mathbf{y}_1^A, \ldots, 1_i^A, \ldots, \mathbf{y}_n^A)$ and $\mathbf{Y}^B = (\mathbf{y}_1^B, \ldots, 0_i^B, \ldots, \mathbf{y}_n^B)$ which are different at the $i$-th position, the data flow can be summarized by (1) and (2). We force the $i$-th latent encoding of $B$ to be zero in order to prevent from trivial solutions as we will discuss in Section 3.3.

The encoder and decoder receive two types of losses: (1) the reconstruction loss,

$$L_{reconstruct} = \|A - A_1\|_1 + \|B - B_1\|_1 \tag{3}$$

which measures the reconstruction quality after a sequence of encoding and decoding; (2) the standard GAN loss,

$$L_{GAN} = -\mathbb{E}[\log(\mathrm{D}(A_2|\mathbf{y}_i^A = 1))] - \mathbb{E}[\log(\mathrm{D}(B_2|\mathbf{y}_i^B = 0))] \tag{4}$$

which measures how realistic the generated images are. The discriminator takes the generated image and the $i$-th element of its label as inputs, and outputs a number which indicates how realistic the input image is. The larger the number is, the more realistic the image is. Omitting the coefficient, the loss function for the encoder and decoder is

$$L_G = L_{reconstruct} + L_{GAN}. \tag{5}$$

The discriminator D receives the standard GAN discriminator loss

$$L_{D_1} = -\mathbb{E}[\log(\mathrm{D}(A|\mathbf{y}_i^A = 1))] - \mathbb{E}[\log(1 - \mathrm{D}(B_2|\mathbf{y}_i^A = 1))] \tag{6}$$

$$L_{D_0} = -\mathbb{E}[\log(\mathrm{D}(B|\mathbf{y}_i^B = 0))] - \mathbb{E}[\log(1 - \mathrm{D}(A_2|\mathbf{y}_i^B = 0))] \tag{7}$$

$$L_D = L_{D_1} + L_{D_0} \tag{8}$$

where $L_{D_1}$ drives D to tell $A$ from $B_2$, and $L_{D_0}$ drives D to tell $B$ from $A_2$.

## 3.3 Annihilating Operation Prevents from Trivial Solutions

Through experiments, we observe that there exist trivial solutions to our model without the annihilating operation. We just take the single-attribute case as an example. Suppose that $\mathrm{Enc}(A) = [a, z_a]$ and $\mathrm{Enc}(B) = [b, z_b]$, we can get four children without annihilating operation

$$A_1 = \mathrm{Dec}([a, z_a]), \quad B_1 = \mathrm{Dec}([b, z_b]), \quad A_2 = \mathrm{Dec}([b, z_a]), \quad B_2 = \mathrm{Dec}([a, z_b]) \tag{9}$$

The reconstruction loss makes it invertible between the latent encoding space and image space. The adversarial discriminator D is supposed to disentangle the attribute from other information by telling whether $A_2$ looks as real as $B$ and $B_2$ looks as real as $A$ or not. As we know that the generative adversarial networks give the best solution when achieving the Nash equilibrium. But without the annihilating operation, information of the whole image could be encoded into the attribute-relevant part, which means

$$\mathrm{Enc}(A) = [a, 0], \quad \mathrm{Enc}(B) = [b, 0] \tag{10}$$

Therefore, we obtain the following four children

$$A_1 = \mathrm{Dec}([a, 0]), \quad B_1 = \mathrm{Dec}([b, 0]), \quad A_2 = \mathrm{Dec}([b, 0]), \quad B_2 = \mathrm{Dec}([a, 0]) \tag{11}$$

In this situation, the discriminator D cannot discriminate $A_2$ from $B$, since they share the same latent encodings. By reconstruction loss, $A_2$ and $B$ are exactly the same image, which is against our expectation that $A_2$ should depict the person from $A$ with the attribute borrowed from $B$. The same thing happens to $B_2$ and $A$ as well.

To prevent from learning trivial solutions, we adopt the annihilating operation by replacing the recessive pattern $b$ with a zero tensor of the same size[1]. If information of the whole image were encoded into the attribute-relevant part, the four children in this case are

$$A_1 = \mathrm{Dec}([a, 0]), \quad B_1 = \mathrm{Dec}([0, 0]), \quad A_2 = \mathrm{Dec}([0, 0]), \quad B_2 = \mathrm{Dec}([a, 0]) \tag{12}$$

The encodings of $B_1$ and $A_2$ contain no information at all, thus neither the person in $B_1$ nor $A_2$ who is supposed to be the same as in $B$ can be reconstructed by Dec. This forces the attribute-irrelevant part to encode some information of images.

---

[1] Use `tf.zeros_like()` in TensorFlow implementation.

### 3.4 ITERATIVE TRAINING

To reduce the difficulty of disentangling multiple attributes, we take the strategy of iterative training: we update our model using a pair of images with opposite labels at a certain position each time. Suppose that we are at the $i$-th position, the label of image $A$ is $(\mathbf{y}_1^A, \ldots, 1_i^A, \ldots, \mathbf{y}_n^A)$, while the label of image $B$ is $(\mathbf{y}_1^B, \ldots, 0_i^B, \ldots, \mathbf{y}_n^B)$. During each iteration, as $i$ goes through from 1 to $n$ repeatedly, our model fed with such a pair of images can disentangle multiple attributes one-by-one.

Compared with training with random pairs of images, iterative training is proved to be more effective. Random pairs of images means randomly selecting pairs of images each time without label constraints. A pair of images with different labels is called a *useful pair*.

We theoretically show that our iterative training is much more efficient than random image pairs especially when the dataset is unbalanced. All proofs can be found in the Appendix.

**Theorem 1.** *Let $\mathscr{X} = \{(\mathbf{X}^1, \mathbf{Y}^1), \ldots, (\mathbf{X}^m, \mathbf{Y}^m)\}$ denote the whole multi-attribute image dataset, where $\mathbf{X}^i$ is a multi-attribute image and its label $\mathbf{Y}^i = (\mathbf{y}_1^i, \ldots, \mathbf{y}_n^i)$ is an $n$-dimensional vector. There are totally $2^n$ kinds of labels, denoted by $\mathscr{L} = \{l_1, \ldots, l_{2^n}\}$. The number of images with label $l_i$ is $m_i$, and $\sum_{i=1}^{2^n} m_i = m$. To select all useful pairs at least once, the expected numbers of iterations needed for randomly selecting pairs and for iterative training are denoted by $\mathrm{E}_1$ and $\mathrm{E}_2$ respectively. Then,*

$$\mathrm{E}_1 = m^2 \left( 1 + \frac{1}{2} + \cdots + \frac{1}{m^2 - \sum_{i=1}^{2^n} m_i^2} \right) \tag{13}$$

$$\mathrm{E}_2 \le 2n \cdot \max_{s=1,\ldots,n} \sum_{i \in I_s, j \in J_s} m_i m_j \left( 1 + \frac{1}{2} + \cdots + \frac{1}{m^2 - \sum_{k_1=1}^{2^{n-1}} (m_{i_{k_1}} + m_{j_{k_1}})^2} \right) \tag{14}$$

*where $I_s$ represents the indices of labels where the $s$-th element is $1$, and $J_s$ represents the indices of labels where the $s$-th element is $0$.*

**Definition 1.** (Balancedness) Define the balancedness of a dataset $\mathscr{X}$ described above with respect to the $s$-th attribute as follows:

$$\rho_s = \frac{\sum_{i \in I_s} m_i}{\sum_{j \in J_s} m_j} \tag{15}$$

where $I_s$ represents the indices of labels where the $s$-th element is $1$, and $J_s$ represents the indices of labels where the $s$-th element is $0$.

**Theorem 2.** *We have $\mathrm{E}_2 \le \mathrm{E}_1$, when*

$$n \le \min_s \frac{(\rho_s + 1)^2}{2\rho_s}. \tag{16}$$

*Specifically, $\mathrm{E}_2 \le \mathrm{E}_1$ holds true for all $n \le 2$.*

The property of the function $(\rho + 1)^2/(2\rho)$ suits well with the definition of balancedness, because it attains the same value for $\rho$ and $1/\rho$, which is invariant to different labeling methods. Its value gets larger as the dataset becomes more unbalanced. The minimum is obtained at $\rho = 1$, which is the case of a balanced dataset.

Theorem 2 demonstrates that the iterative training mechanism is always more efficient than random pairs of images when the number of attributes met the criterion (16). As the dataset becomes more unbalanced, $(\rho_s + 1)^2/(2\rho_s)$ goes larger, which means (16) can be more easily satisfied. More importantly, iterative training helps stabilize the training process on unbalanced datasets. For example, given a two-attribute dataset, the number of data of each kind is as follows:

Table 1: The example of an unbalanced two-attribute dataset.

| Label | $(0,0)$ | $(0,1)$ | $(1,0)$ | $(1,1)$ |
|---|---|---|---|---|
| Number of data | 1 | 1 | $m$ | $m$ |

If $m \gg 1$ is a very large number, then it is highly likely that we will select a pair of images whose labels are $(1,0)$ and $(1,1)$ each time by randomly selecting pairs. We ignore the pair of images

$$A \qquad B \qquad A_2 \qquad B_2 \qquad A_1 \qquad B_1$$

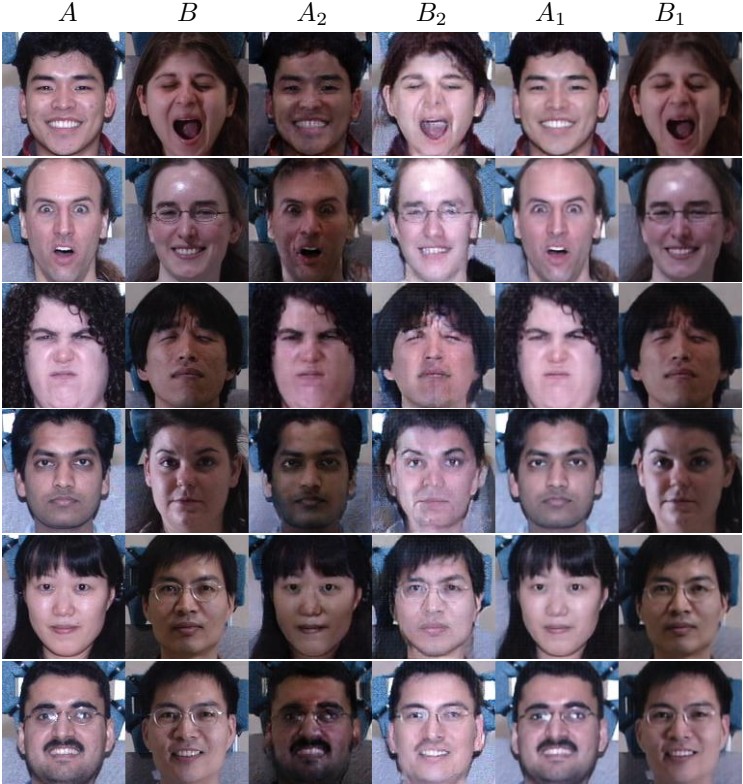

Figure 2: Manipulating illumination factors on the Multi-PIE dataset. From left to right, the six images in a row are: original images $A$ with light illumination and $B$ with the dark illumination, newly generated images $A_2$ and $B_2$ by swapping the illumination-relevant piece in disentangled representations, and reconstructed images $A_1$ and $B_1$.

whose labels are $(1,0)$ and $(1,0)$ or $(1,1)$ and $(1,1)$, though these two cases have equal probabilities of being chosen. Because they are not useful pairs, thus do not participated in training. In this case, most of the time the model is trained with respect to the second attribute, which will cause the final learnt model less effective to the first attribute. However, iterative training can prevent this from happening, since we update our model evenly with respect to two attributes.

# 4 EXPERIMENTS

In this section, we perform different kinds of experiments on two real-world datasets to validate the effectiveness of our methods. We use the RMSProp (Sutskever et al., 2013) optimization method initialized by a learning rate of 5e-5 and momentum 0. All neural networks are equipped with Batch Normalization (Ioffe & Szegedy, 2015) after convolutions or deconvolutions. We used Leaky Relu (Maas et al., 2013) as the activation function in the encoder. Besides, we adopt strategies mentioned in Wasserstein GAN (Arjovsky et al., 2017) for stable training. More details will be available online. We divide all images into training images and test images according to the ratio of 9:1. All of the following results are from test images without cherry-picking.

## 4.1 MULTI-PIE DATABASE

The Multi-PIE (Gross et al., 2010) face database contains over 750,000 images of 337 subjects captured under 15 view points and 19 illumination conditions. We collecte all front faces images of different illuminations and align them based on 5-point landmarks on eyes, nose and mouth. All aligned images are resized into $128 \times 128$ as inputs in our experiments. We label the light illumination face images by 1 and the dark illumination face images by 0. As shown in Figure 2,

the illumination on one face is successfully transferred into the other face without modifying any other information in the images. This demonstrates that DNA-GAN can effectively disentangle the illumination factor from other factors in the latent space.

## 4.2 CELEBA DATASET

CelebA (Liu et al., 2015) is a dataset composed of 202599 face images and 40 attribute binary vectors and 5 landmark locations. We use the aligned and cropped version and scaled all images down to $64 \times 64$. To better demonstrate the advantage of our method, we choose TD-GAN (Wang et al., 2017) and IcGAN (Perarnau et al., 2016) for comparisons.

As we mentioned before, TD-GAN requires the explicit id information in the label, thus cannot be applied to the CelebA dataset directly. To overcome this limitation, we use some channels to encode the id information in its latent representations. In our experiments, the id information is preserved when swapping the attribute information in the latent encodings. We also compared the experimental results of IcGAN with ours in the celebA dataset. The following results are obtained using the the official code and pre-trained celebA model provided by the author[2].

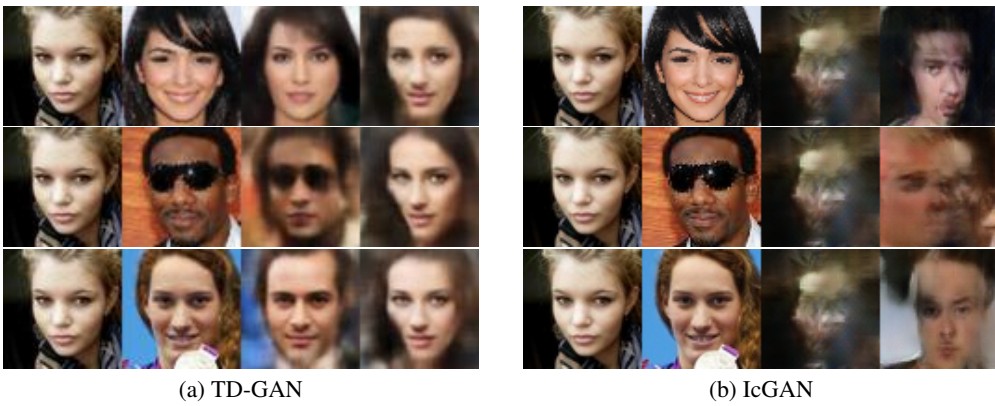

(a) TD-GAN            (b) IcGAN

Figure 3: The experimental results of TD-GAN and IcGAN on CelebA dataset. Three rows indicates the swapping attributes of Bangs, Eyeglasses and Smiling. For each model, the four images in a row are: two original images, and two newly generated images by swapping the attributes. The third image is generated by adding the attribute to the first one, and the fourth image is generated by removing the attribute from the second one.

As displayed in Figure 3a, modified TD-GAN encounters the problem of trivial solutions. Without id information explicitly contained in the label, TD-GAN encodes the information of the whole image into the attribute-related part in the latent representations. As a result, two faces are swapped directly. Whereas in Figure 3b, the quality of images generated by IcGAN are very bad, which is probably due to the multi-stage training process of IcGAN. Since the overall effect of the model relies much on the each stage.

DNA-GAN is able to disentangle multiple attributes in the latent representations as shown in Figure 4. Since different attributes are encoded in different DNA pieces in our latent representations, we are able to interpolate the attribute subspaces by linear combination of disentangled encodings. Figure 4a, 4b and 4c present disentangled attribute subspaces spanned by any two attributes of Bangs, Eyeglasses and Smiling. They demonstrate that our model is effective in learning disentangled representations. Figure 4d shows the hairstyle transfer process among different Bangs styles. It is worth mentioning that the top-left image in Figure 4d is outside the CelebA dataset, which further validate the generalization potential of our model on unseen data. Please refer to Figure 5 in the Appendix for more results.

---

[2]https://github.com/Guim3/IcGAN

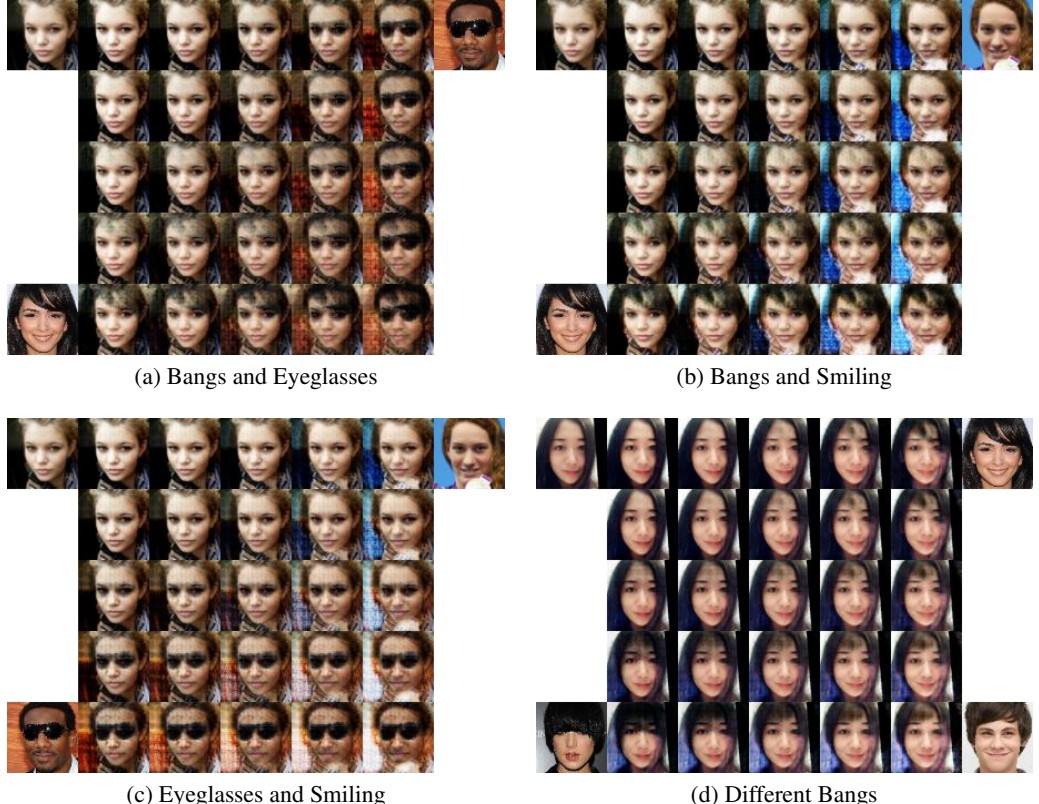

| (a) Bangs and Eyeglasses | (b) Bangs and Smiling |
|---|---|

| (c) Eyeglasses and Smiling | (d) Different Bangs |
|---|---|

Figure 4: The interpolation results of DNA-GAN. Figure 4a, 4b and 4c display the disentangled attribute subspaces spanned by any two attributes of Bangs, Eyeglasses and Smiling. Figure 4d shows the attribute subspaces spanned by several Bangs feature vectors. Besides, the top-left image in Figure 4d is outside the CelebA dataset.

## 5 CONCLUSION

In this paper, we propose a supervised algorithm called DNA-GAN that can learn disentangled representations from multi-attribute images. The latent representations of images are DNA-like, consisting of attribute-relevant and attribute-irrelevant parts. By the annihilating operation and attribute hybridization, we are able to create new latent representations which could be decoded into novel images with designed attributes. The iterative training strategy effectively overcomes the difficulty of training on unbalanced datasets and helps disentangle multiple attributes in the latent space.

The experimental results not only demonstrate that DNA-GAN is effective in learning disentangled representations and image editing, but also point out its potential in interpretable deep learning, image understanding and transfer learning.

There also exist some limitations of our model. Without strong guidance on the attribute-irrelevant parts, some background information is encoded into the attribute-relevant part. As we can see in Figure 4, the background color gets changed when swapping attributes. Besides, our model may fail when several attributes are highly correlated with each other. For example, Male and Mustache are statistically dependent, which are hard to disentangle in the latent representations. These are left as our future work.

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

APPENDIX

To prove Theorem 1, we need the following lemma.

**Lemma 1.** *A set $S = \{s_1, \ldots, s_m\}$ has $m$ different elements, from which elements are being selected equally likely with replacement. The expected number of trials needed to collect a subset $R = \{s_1, \ldots, s_n\}$ of $n (1 \leq n \leq m)$ elements is*

$$m \cdot \left( \frac{1}{1} + \frac{1}{2} + \cdots + \frac{1}{n} \right).$$

*Proof.* Let $T$ be the time to collect all $n$ elements in the subset $R$, and let $t_i$ be the time to collect the $i$-th new elements after $i-1$ elements in $R$ have been collected. Observe that the probability of collecting a new element is $p_i = (n - (i-1))/m$. Therefore, $t_i$ is a geometrically distributed random variable with expectation $1/p_i$. By the linearity of expectations, we have:

$$\begin{aligned}
\mathbb{E}(T) &= \mathbb{E}(t_1) + \mathbb{E}(t_2) + \cdots + \mathbb{E}(t_n) \\
&= \frac{1}{p_1} + \frac{1}{p_2} + \cdots + \frac{1}{p_n} \\
&= \frac{m}{n} + \frac{m}{n-1} + \cdots + \frac{m}{1} \\
&= m \cdot \left( \frac{1}{1} + \frac{1}{2} + \cdots + \frac{1}{n} \right).
\end{aligned}$$

$\square$

*Proof.* (of Theorem 1)

We first consider the case of randomly selecting pairs. All possible image pairs are actually in the product space $\mathscr{X} \times \mathscr{X}$, whose cardinality is $m^2$. If we take the order of two images in a pair into consideration, the number of possible pairs is $m^2$. Recall that the *useful pair* denotes a pair of image of different labels. Therefore, the number of all useful pairs is $\sum_{i \neq j} m_i m_j$. By Lemma 1, the expected number of iterations for randomly selecting pairs to select all useful pairs at least once is

$$\begin{aligned}
\mathrm{E}_1 &= m^2 \left( 1 + \frac{1}{2} + \cdots + \frac{1}{\sum_{i \neq j} m_i m_j} \right) \\
&= m^2 \left( 1 + \frac{1}{2} + \cdots + \frac{1}{\sum_{i=1}^{2^n} (m_i \sum_{j \neq i} m_j)} \right) \\
&= m^2 \left( 1 + \frac{1}{2} + \cdots + \frac{1}{\sum_{i=1}^{2^n} m_i (m - m_i)} \right) \\
&= m^2 \left( 1 + \frac{1}{2} + \cdots + \frac{1}{m^2 - \sum_{i=1}^{2^n} m_i^2} \right). \qquad (17)
\end{aligned}$$

Now we consider the case of iterative training. We always select a pair of images of different labels each time. Suppose we are selecting images with opposite labels at the $s$-th position. Let $I_s$ denote the indices of all labels with the $s$-th element 1, and $J_s$ denote the indices of all labels with the $s$-th element 0, where $|I_s| = |J_s| = 2^{n-1}$. Then we consider the subproblem by neglecting the first position in data labels, the number of all possible pairs is $2 \sum_{i \in I_s, j \in J_s} m_i m_j$ (regarding of order),

and the number of useful pairs is

$$\sum_{k_1 \neq k_2} (m_{i_{k_1}} + m_{j_{k_1}})(m_{i_{k_2}} + m_{j_{k_2}})$$

$$= \sum_{k_1=1}^{2^{n-1}} \sum_{k_2 \neq k_1} (m_{i_{k_1}} + m_{j_{k_1}})(m_{i_{k_2}} + m_{j_{k_2}})$$

$$= \sum_{k_1=1}^{2^{n-1}} (m_{i_{k_1}} + m_{j_{k_1}})(m - m_{i_{k_1}} - m_{j_{k_1}})$$

$$= m^2 - \sum_{k_1=1}^{2^{n-1}} (m_{i_{k_1}} + m_{j_{k_1}})^2. \tag{18}$$

Therefore, the expectation to select all useful pairs at least once regardless of the $s$-th element in the label is

$$E_{\backslash s} = 2 \sum_{i \in I_s, j \in J_s} m_i m_j \left( 1 + \frac{1}{2} + \cdots + \frac{1}{m^2 - \sum_{k_1=1}^{2^{n-1}} (m_{i_{k_1}} + m_{j_{k_1}})^2} \right) \tag{19}$$

Since we rotate the subscript $s$ from 1 to $n$, the expected number of iterations for iterative training to select all useful pairs at least once is

$$E_2 \leq n \cdot \max_{s=1,\ldots,n} E_{\backslash s}$$

$$= 2n \cdot \max_{s=1,\ldots,n} \sum_{i \in I_s, j \in J_s} m_i m_j \left( 1 + \frac{1}{2} + \cdots + \frac{1}{m^2 - \sum_{k_1=1}^{2^{n-1}} (m_{i_{k_1}} + m_{j_{k_1}})^2} \right). \tag{20}$$

$\square$

*Proof.* (of Theorem 2) We firstly show that

$$\sum_{k_1=1}^{2^{n-1}} (m_{i_{k_1}} + m_{j_{k_1}})^2 \geq \sum_{k_1=1}^{2^{n-1}} (m_{i_{k_1}}^2 + m_{j_{k_1}}^2) = \sum_{i=1}^{2^n} m_i^2 \tag{21}$$

According to the result of Theorem 1 and the Definition 1 of balancedness, we have

$$E_2 = 2n \cdot \max_s \sum_{i \in I_s, j \in J_s} m_i m_j \left( 1 + \frac{1}{2} + \cdots + \frac{1}{m^2 - \sum_{k_1=1}^{2^{n-1}} (m_{i_{k_1}} + m_{j_{k_1}})^2} \right)$$

$$\leq 2n \cdot \max_s \sum_{i \in I_s, j \in J_s} m_i m_j \left( 1 + \frac{1}{2} + \cdots + \frac{1}{m^2 - \sum_{i=1}^{2^n} m_i^2} \right)$$

$$= 2n \cdot \max_s \left( \sum_{i \in I_s} m_i \right) \left( \sum_{j \in J_s} m_j \right) \left( 1 + \frac{1}{2} + \cdots + \frac{1}{m^2 - \sum_{i=1}^{2^n} m_i^2} \right)$$

$$= 2n \cdot \max_s \frac{\rho_s m}{\rho_s + 1} \frac{m}{\rho_s + 1} \left( 1 + \frac{1}{2} + \cdots + \frac{1}{m^2 - \sum_{i=1}^{2^n} m_i^2} \right)$$

$$= \max_s \frac{2n\rho_s}{(\rho_s + 1)^2} \cdot m^2 \left( 1 + \frac{1}{2} + \cdots + \frac{1}{m^2 - \sum_{i=1}^{2^n} m_i^2} \right)$$

$$\leq E_1. \tag{22}$$

Specifically, if $n \leq 2$,

$$\frac{2n\rho_s}{(\rho_s + 1)^2} \leq \frac{4\rho_s}{(\rho_s + 1)^2} \leq 1. \tag{23}$$

The inequality holds true forever.

$\square$

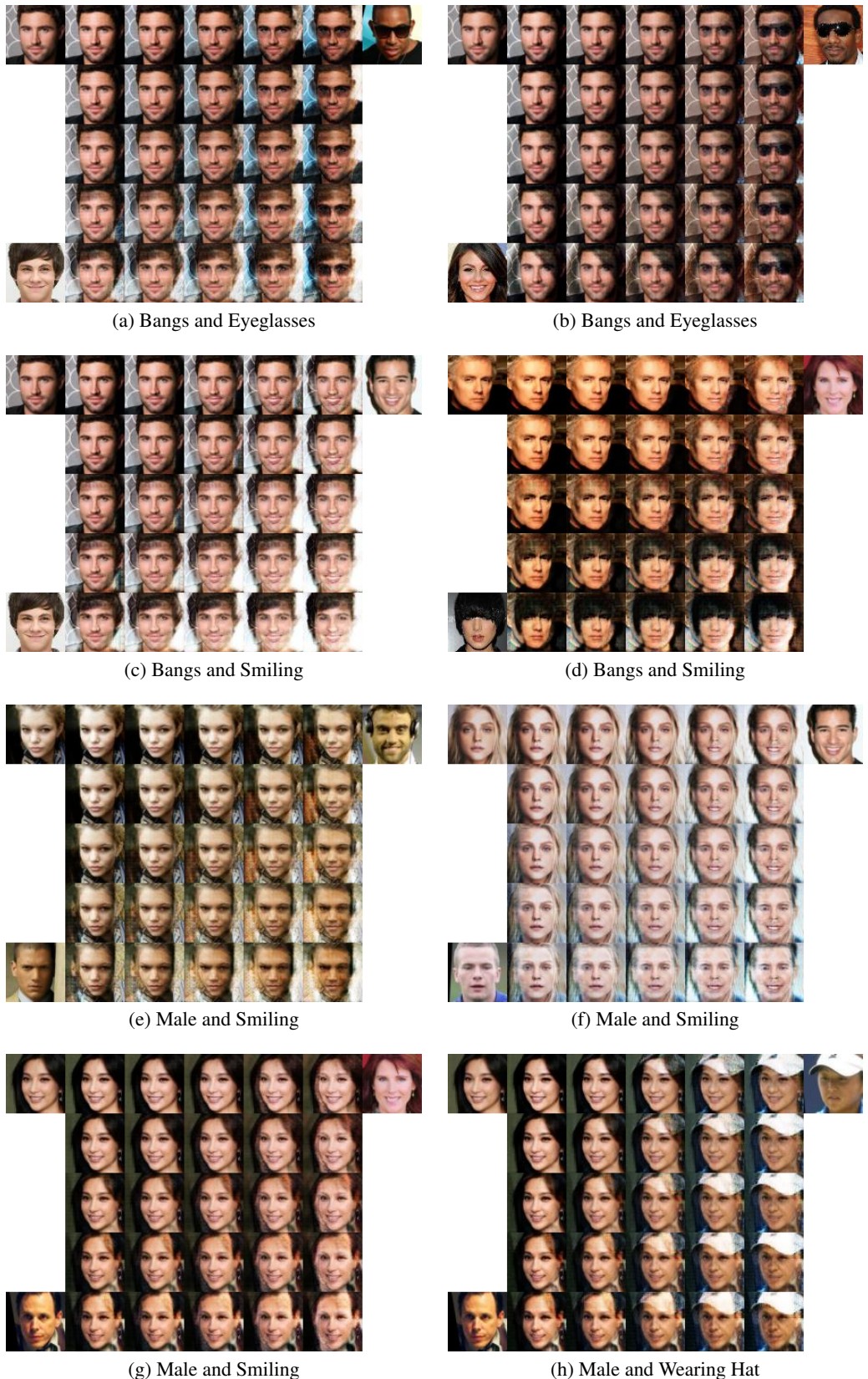

(a) Bangs and Eyeglasses

(b) Bangs and Eyeglasses

(c) Bangs and Smiling

(d) Bangs and Smiling

(e) Male and Smiling

(f) Male and Smiling

(g) Male and Smiling

(h) Male and Wearing Hat

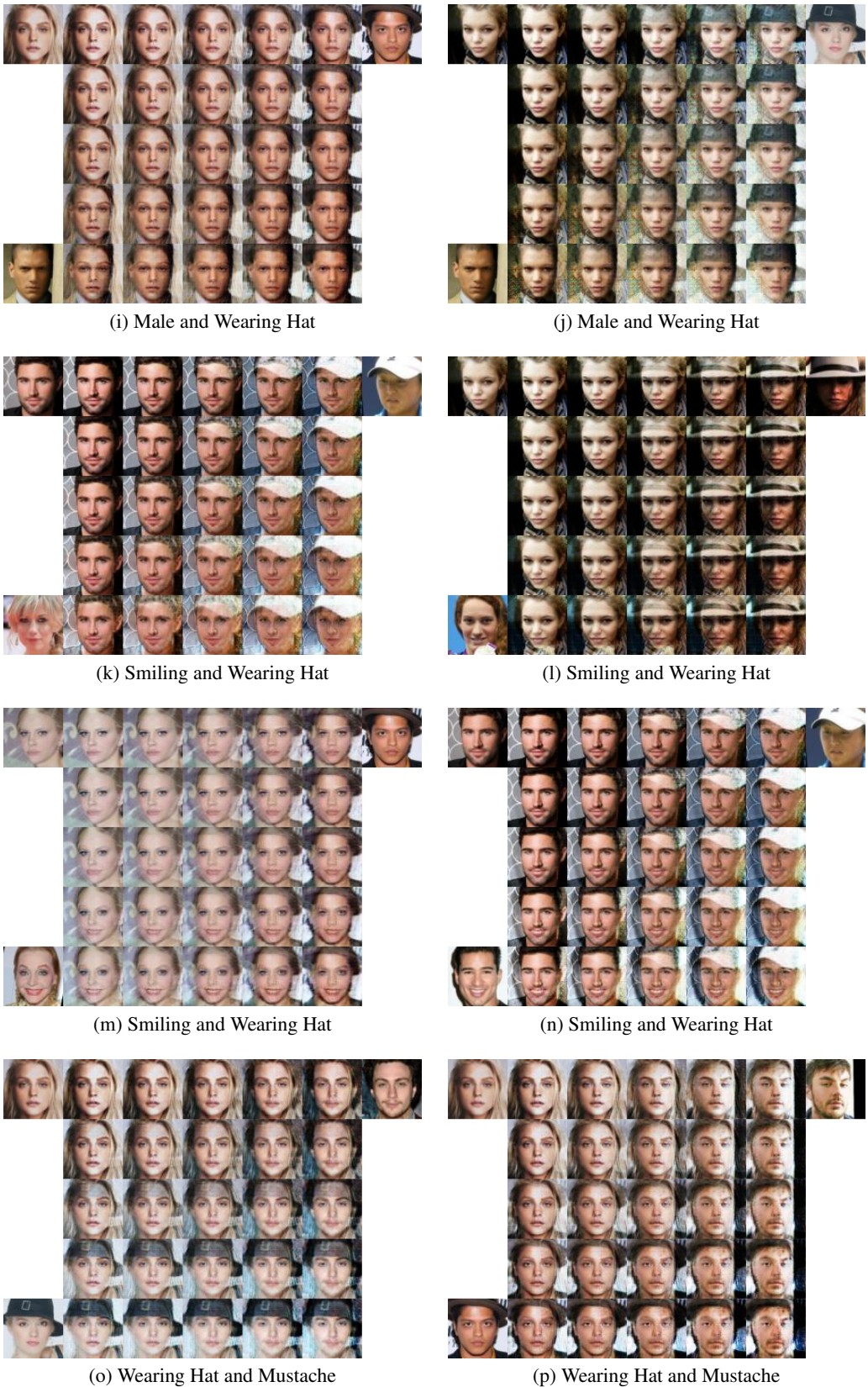

(i) Male and Wearing Hat

(j) Male and Wearing Hat

(k) Smiling and Wearing Hat

(l) Smiling and Wearing Hat

(m) Smiling and Wearing Hat

(n) Smiling and Wearing Hat

(o) Wearing Hat and Mustache

(p) Wearing Hat and Mustache

Figure 5: More experimental results of DNA-GAN.

