# OpenReview forum: "DNA-GAN: Learning Disentangled Representations from Multi-Attribute Images"
_ICLR.cc/2018/Conference — Invite to Workshop Track_

### Official Review · AnonReviewer1 · 2017-11-25
**Interesting idea but unclear description of the method and not very convincing results**

**Rating:** 4
**Confidence:** 4

**Review:**

This paper proposes to disentangle attributes by forcing a representation where individual components of this representation account for individual attributes.

Pros:
+ The idea of forcing different parts of the latent representation to be responsible for different attributes appears novel.
+ A theoretical guarantee of the efficiency of an aspect of the proposed method is given.

Cons:
- The results are not very appealing visually. The results from the proposed method do not seem much better than the baselines. What is the objective for the images in Fig. 4? For example I'm looking at the bottom right, and that image looks more like a merger of images, than a modification of the image in the top-left but adding the attributes of choice.
- Quantitative results are missing.
- Some unclarity in the description of the method; see below.

Questions/other:
- What is meant by "implicit" models? By "do not anchor a specific meaning into the disentanglement"? By "circumscribed in two image domains"?
- Why does the method require two images?
- In the case of images, what is a dominant vs recessive pattern?
- It seems artificial to enforce that "the attribute-irrelevant part [should] encode some information of images".
- Why are (1, 0) and (1, 1) not useful pairs?
- Need to be more specific: "use some channels to encode the id information".

---

> ### Author Response · Authors · 2017-12-02
> **Response to AnonReviewer1**
>
> Thanks for your feedbacks.
>
> 1. In Fig. 4,  the right bottom image was generated from the top left image with two attributes from bottom left and top right images. Figure. 3 displays the baseline of TD-GAN and IcGAN in one-attribute case. The left two columns are original images. Since TD-GAN encountered the problem of trivial solutions and IcGAN cannot generate real-looking images in single-attribute case, so we did not show the their results in the multi-attribute case. Actually, the multi-attribute case is much harder than the single-attribute case. Actually the visual effect is not bad. (Please see Figure 2.) if you would like to compare results of celebA on the 64*64 resolution level, please look at VAE-GAN (https://github.com/anitan0925/vaegan).
>
> An important factor that renders the unfairness of comparison is DNA-GNA is able to do image generation by exemplar, which is much more difficult than many other models. As shown in Figure 5,  (j) and (l) display two different kinds of generated hats at top of the same person. Many other models are not able to add different hats to the same image. This is what I mean image generation by exemplar.
>
> Moreover, the overall focus of our paper is not image generation. The visual effect can be improved by extensive hyper-parameter tuning, but these figures are used to demonstrate that our model can learn disentangled representations in our latent representations. The contribution of a novel learning disentangled representations using weakly labeled multi-image should be cared about. The iterative training strategy addresses the problem of training on unbalanced dataset and improves the training efficiency. The annihilating operation makes the image generation by exemplar works. Theoretical connection between the training efficiency and the balancedness was given. These are the focus of our paper.
>
> 2. There is no reasonable quantitative measure for GAN related papers. So we did not tune our parameters heavily as many other papers did. The figures in our paper were used to demonstrate that multiple attributes were indeed disentangled in our latent representations. The visual effect can be improved better, but it is not the focus of our paper. We care more about the advantage of our model in overcoming difficulties existing in other models, such as 1) difficulty of training on unbalanced multi-attribute datasets, 2) trivial solutions without preserving id information in other methods of image generation by exemplar 3) training on weakly labeled dataset.
>
> 3. "Implicit models" means the probability distribution of training samples can not be explicitly formulated.
>
> "do not anchor a specific meaning into the disentanglement" means: we cannot predict what factors of variation beforehand in unsupervised methods.
>
> "circumscribed in two image domains" means CycleGAN, DTN, UNIT, GeneGAN are only able to do image translation between two image domains with respect to one attribute. However, our model are able to do multi-attribute image generation.
>
> 4. We need two images with different attributes at i-th position each time. In our model, we swapped the latent encodings to generate novel crossbreeds, which can be decoded into novel images with new attributes.
>
> 5. Dominant pattern means the i-th label is 1, while the recessive pattern means the i-th label is 0.
>
> 6. The attribute-irrelevant part z_a or z_b is left for encoding information of background or image identity. Because the attribute-related parts are only able to represent part of the image information. For example, in three-attribute case, [Bangs, Eyeglasses, Smiling], there are many other information in images except for Bangs, Eyeglasses and Smiling, such as wearing hat, mustache, the person identity and background.
>
> 7. (1, 0) and (1, 1) are useful pairs. The sentence below Fig. 2 "Because they are not useful pairs, thus do not participated in training"  means  {(1, 0) and (1, 0)} or {(1, 1) and (1, 1)} is not useful pairs.
>
> 8. The id information was similarly encoded in z. The latent representations are 4-d tensor, each dimension of which represents batch size, height, width, channel. We divide some channels to encode the id information.

---

> > ### Comment · AnonReviewer1 · 2018-01-13
> > **thanks**
> >
> > Thanks for the rebuttal. It clears up some confusion but my score remains slightly negative.

---

### Official Review · AnonReviewer3 · 2017-11-26
**Review summary "DNA-GAN: Learning Disentangled Representations from Multi-Attribute Images"**

**Rating:** 5
**Confidence:** 4

**Review:**

Summary:
This paper investigated the problem of attribute-conditioned image generation using generative adversarial networks. More specifically, the paper proposed to generate images from attribute and latent code as high-level representation. To learn the mapping from image to high-level representations, an auxiliary encoder was introduced. The model was trained using a combination of reconstruction (auto-encoding) and adversarial loss. To further encourage effective disentangling (against trivial solution), an annihilating operation was proposed together with the proposed training pipeline. Experimental evaluations were conducted on standard face image databases such as Multi-PIE and CelebA.

== Novelty and Significance ==
Multi-attribute image generation is an interesting task but has been explored to some extent. The integration of generative adversarial networks with auto-encoding loss is not really a novel contribution.
-- Autoencoding beyond pixels using a learned similarity metric. Larsen et al., In ICML 2016.

== Technical Quality ==
First, it is not clear how was the proposed annihilating operation used in the experiments (there is no explanation in the experimental section). Based on my understanding, additional loss was added to encourage effective disentangling (prevent trivial solution). I would appreciate the authors to elaborate this a bit.

Second, the iterative training (section 3.4) is not a novel contribution since it was explored in the literature before (e.g., Inverse Graphics network). The proof developed in the paper provides some theoretical analysis but cannot be considered as a significant contribution.

Third, it seems that the proposed multi-attribute generation pipeline works for binary attribute only. However, such assumption limits the generality of the work. Since the title is quite general, I would assume to see the results (1) on datasets with real-valued attributes, mixture attributes or even relative attributes and (2) not specific to face images.
-- Learning to generate chairs with convolutional neural networks. Dosovitskiy et al., In CVPR 2015.
-- Deep Convolutional Inverse Graphics Network. Kulkarni et al., In NIPS 2015.
-- Attribute2Image: Conditional Image Generation from Visual Attributes. Yan et al., In ECCV 2016.
-- InfoGAN: Interpretable Representation Learning by Information Maximizing Generative Adversarial Nets. Chen et al., In NIPS 2016.

Additionally, considering the generation quality, the CelebA samples in the paper are not the state-of-the-art. I suspect the proposed method only works in a more constrained setting (such as Multi-PIE where the images are all well aligned).

Overall, I feel that the submitted version is not ready for publication in the current form.

---

> ### Author Response · Authors · 2017-12-02
> **Response to AnonReviewer3**
>
> Thanks for your feedback.
>
> 1. There are many model structures that integrate of generative adversarial networks with auto-encoding loss, such CycleGAN, DTN, UNIT, etc.  But this is not our contribution. Instead, the key point is to learn the disentangled representations by iteratively swapping the attribute of two images. This idea can help address the problem of underdetermined attribute pattern. For example, we want to generate a facial images with the particular eyeglasses in another image, not a general eyeglasses. Many existing methods are only able to add one kind of eyeglasses to a certain image. But our method can add various kinds of eyeglasses by swapping the attribute-part in latent encodings. Besides, it is not easy to make this idea work,  we need annihilating operation to prevent from trivial solutions.
>
> - CycleGAN: Unpaired Image-to-Image Translation using Cycle-Consistent Adversarial Networks
> - DTN: Unsupervised cross-domain image generation
> - UNIT: Unsupervised image-to-image translation networks
>
> We will add VAE-GAN in our reference.
>
> 2. As we explained in Section 3.3, the annihilating operation is to replace a tensor by a zero tensor of the same size.
> The footnote 1 also explains the tensorflow implementation: tf.zeros_like(). No additional loss is necessary. Section 3.3 gives a simple example to illustrate why the solution would be trivial without the annihilating operation. I would appreciate that you read Section 3.3 for details.
>
> The comparison experiments with TD-GAN demonstrate the importance of the annihilating operation. Without it, TD-GAN encodes all information into the attribute part. As a consequence,  two original images rather than the attribute get swapped.
>
> 3.  The motivations of iterative training comes from the difficulty of training on the unbalanced dataset. The iterative training strategy was employed to overcome this difficulty and increase the training efficiency. Besides, the theoretical parts pointed out the close connection between training efficiency and the balancedness of dataset.  These were not explored in the previous literature.
>
> 4. The current pipeline is indeed only for binary attribute. But the requirement for weakly supervised label 0/1 is an advantage to some extent. In our experiments on the MultiPie dataset, the illumination factor was only labeled for dark (0) to light (1), but our model can interpolate the illumination ranging from dark (0) to light (1), which is a real-value. Most image datasets are discretely labeled, therefore I believe our model can further apply to many other datasets with cheap expense of labeling (0/1). Of course many unsupervised methods are naturally suitable for this case, since they do not need label. But we cannot predict what factors of variation beforehand. Instead, we make up stories after they work.
>
> 5. There is no reasonable quantitative measure for GAN related papers. So we did not tune our parameters heavily as many other papers did. The figures displayed in the paper are the initial successful results. What we cares is a generally effective method. I believe our model can achieve very impressive results if more machines and efforts being devoted.  I don't want to select very good pictures as IcGAN did but totally useless in practice. Besides, do remember that our model can do image generation by exemplars, rather than simply adding mean attributes to images. This is particular difficult in the multi-attribute case. As far as I know, many other methods are not able to to this. (e.g. TD-GAN needs the labeled id information  when swapping the attribute)

---

> > ### Comment · AnonReviewer3 · 2017-12-02
> > **Questions regarding author feedback**
> >
> > Dear authors,
> >
> > Thank you for your feedback!
> >
> > 1. "Many existing methods are only able to add one kind of eyeglasses to a certain image. But our method can add various kinds of eyeglasses by swapping the attribute-part in latent encodings."
> > I am not convinced by the argument made here.
> > -- First, can you link me to the "many existing methods" you referred to?
> > CycleGAN is an exception (I don't think it is generative model since no stochastity is involved).
> > Both DTN and UNIT can generate diverse-looking samples.
> > -- Second, I couldn't find evidence in the paper that the proposed method has the capacity to generate diverse-looking eyeglasses.
> >
> > 2. Thank you for the explanation. But I don't see much novelty from the proposed annihilating (it is basically augmenting the sampling distribution that discourages trivial solution).
> >
> > 3. "These were not explored in the previous literature. "
> > I agree the theoretical analysis is your contribution but I don't quite agree with the argument made here. Can you possibly summarize the differences against Inverse Graphics Networks?
> >
> > 4. When presenting your work, please make it crystal clear you are targeting at face images with binary attributes.
> >
> > 5. Current results are not very convincing. Please improve the current form if you think your results are preliminary (not ready for publication). Another suggestion is to demonstrate your approach on more challenging datasets.

---

> > > ### Author Response · Authors · 2017-12-02
> > > **Response**
> > >
> > > Thanks for your feedback!
> > >
> > > 1. For example, in Figure 5,  (j) and (l) display two different kinds of generated hats at top of the same person. However, CycleGAN, DTN and UNIT are not able to generate diverse hats images given the same input image. This is because the attribute information is disentangled from input images in our model.
> > >
> > > 2. The word we used is annihilating not annealing.  It is not related to simulated annealing. I would like to explain the annihilating operation again: replacing the tensor b_i with tf.zeros_like(b_i).
> > > This operation is necessary for the success of image generation by exemplar. Directly swapping attribute part a_i and b_i would cause the network converge to trivial solutions. The failure case of TD-GAN is a good example.
> > >
> > > 3.  DC-IGN randomly selects an active attribute and feeds the other attributes by the average in a mini-batch each time; the iterative training in DNA-GAN is: each attribute was repeatedly selected to be the active attribute and useful pairs are fed for training.
> > >
> > > The differences is: in our model, training with random pairs can be viewed as randomly selecting an active attribute, because the active attribute was chosen according to the different position in two images' labels. This is theoretically proved to be less effective than the iterative training with useful pairs. Besides, we do not need to feed other attributes by the average.
> > >
> > > 4. Thanks for your advice.
> > >
> > > 5. The figures in our paper were used to demonstrate that multiple attributes were indeed disentangled in our latent representations. The visual effect can be improved better, but it is not the focus of our paper. (That should be the focus of this paper. https://openreview.net/forum?id=Hk99zCeAb&noteId=Hk99zCeAb)  We should realize that no single model is perfect in any case by no free lunch theorem. But we should care more about the advantage of every model in overcoming the difficulties in other models as well as its limitation. Our paper addressed 1) difficulty of training on unbalanced multi-attribute datasets, 2) trivial solutions without preserving id information in other methods of image generation by exemplar 3) training on weakly labeled dataset.  Of course, I would like show better results in the modified version later.

---

> > > > ### Comment · AnonReviewer3 · 2017-12-03
> > > > **Sample diversity**
> > > >
> > > > Your answer: "Many existing methods are only able to add one kind of EYEGLASSES to a certain image. But our method can add various kinds of EYEGLASSES by swapping the attribute-part in latent encodings."
> > > >
> > > > My question: "I couldn't find evidence in the paper that the proposed method has the capacity to generate diverse-looking EYEGLASSES."
> > > >
> > > > Your answer: "in Figure 5,  (j) and (l) display two different kinds of generated HATS at top of the same person."
> > > >
> > > > If you want to claim something in your paper and rebuttal, please make sure it is (1) accurate and (2) concrete.
> > > >
> > > > Regarding "However, CycleGAN, DTN and UNIT are not able to generate diverse hats", I am quite skeptical about your comment. If you really want to make such argument, please provide both qualitative and quantitative analysis.

---

> > > > > ### Author Response · Authors · 2017-12-03
> > > > > **Response**
> > > > >
> > > > > Thanks for your feedback.
> > > > >
> > > > > I appreciate for your rigorous argument. The Hat example I mentioned before was to make you aware of the difference between our method and other method, though it was not elaborate in our original paper. Because I thought the idea and structure of DNA-GAN is naturally distinct from others, thus it is not necessary to point out that. Please do not have a  glimpse of the framework (Fig. 1), and find there is an encoder, a decoder and a discriminator, and say 'Oh, it is nothing new to me'. There are many models that used them, but the idea and detail of each model is totally different.
> > > > >
> > > > > All in all, I wish my previous comments could help correct your misunderstandings towards our paper.  I hope that you read our paper again and evaluate our contribution and originality.

---

### Official Review · AnonReviewer2 · 2017-11-27
**This paper proposed a new method to disentangle different attributes of images.  A novel DNA structure GAN is proposed to manipulate attributes in images.**

**Rating:** 6
**Confidence:** 5

**Review:**

Pros:
1. A new DNA structure GAN is utilized to manipulate/disentangle attributes.

2. Non attribute part (Z) is explicitly modeled in the framework.

3. Based on the experiment results, this proposed method outperformed previous methods (TD-GAN, IcGAN).

Cons:
1. It assumes that each individual piece represents an independent factor of variation, which can not hold all the time. The authors also admit that when two factors are dependent, this method might fail.

2. In Lreconstruct, only min difference between A and A1 is considered. How about A and A2 here? It seems that A2 should also be similar with A since only one bit in A2 and A1 is different.

3. Only one attribute can be "manipulated" each time? Is it possible to change more than one attribute each time in this method?

---

> ### Author Response · Authors · 2017-12-02
> **Response to AnonReviewer2**
>
> Thanks for your review and comments.
>
> 1. When two factors are statistically dependent with each other, many similar methods would fail, either. For example, considering two attribute male and mustache, they appears or disappears almost simultaneously since they statistically dependent with each other. The model would consider them as on attribute. Anyway, it is a fundamental and hard problem in disentangled representation learning.
>
> 2.  In our model framework, A2 should display the person from A without the i-th attribute a_i, and B2 should display the person from B with the i-th attribute a_i. We cannot enforce reconstruction loss between A2 and A, because they are the same person with different attribute. Imaging that A is a person with eyeglasses and A2 should be the person without eyeglasses, it is not reasonable to enforce the reconstruction loss between them, since they looks different.
>
> 3. In the training process, we only need to swap only one attribute each time. By iterative training, DNA-GAN could disentangle multiple attributes. If we change two or more attributes in the training process, then the number combination of all attributes would be exponentially large. For example, if we have three attributes in total, then the number of combinations is 2^3-1=7. Then the training process would become inefficient. This is why we adopt the strategy of iterative training, which is theoretically proved to be better.  Of course, DNA-GAN could manipulate multiple attributes in the test phase, as shown in Fig. 4 and Fig. 5.

---

### Public Comment · (anonymous) · 2017-11-23
**related work**

The Fader Network architecture also deals with the learning of disentangled representations on multi-attribute images:
 https://arxiv.org/abs/1706.00409 . It is probably a relevant paper to cite, and would provide a better comparison than IcGAN.

---

> ### Author Response · Authors · 2017-11-29
> **Cannot reproduce the experimental results of Fader Networks**
>
> Thanks for pointing out that paper. We tried reproducing the Fader-Networks, however failed to generate real-looking images as shown in the paper. The images were blurry generally even with extensive hyper-parameter tuning. Sometimes it failed on one attribute when training with respect to two attributes. Besides, we have noticed that several reproduction available in github are not able to reproduce the results, either. e.g. https://github.com/hjweide/fader-networks,  https://github.com/hardikbansal/Fader-Networks-Tensorflow.
>
> We would like to cite that paper if the authors could release their official codes. For your information, DNA-GAN works even though we did not carefully select hyper-parameters in our experiment. I believe the visual results could get better with extensive hyper-parameter tuning.

---

### Decision · Program_Chairs · 2018-01-29
**ICLR 2018 Conference Acceptance Decision**

**Decision:**

Invite to Workshop Track

**Comment:**

The method proposed in the paper for latent disentanglement and attribute-conditional image generation is novel to the best of my understanding but reviewers (Anon1 and Anon3) have expressed concerns on the quality of results (CelebA images) as well as on the technical presentation and claims in the paper.

Given the novelty of the proposed method, I would *not* like to recommend a "reject" for this paper but the concerns raised by the reviewers on the quality of results and lack of quantitative results seem valid. Authors rule out possibility of any quantitative results in their response but I am not fully convinced -- in particular, effectiveness of attribute-conditional image generation can be captured by first training an attribute classifier on the generated images and then measuring how often the predicted attributes are flipped when conditioning signal is changed. There are also other metrics in the literature for evaluating generative models.

I would recommend inviting it to the workshop track, given that the work is novel and interesting but has scope for improvements.